# Implementation of Detection System for Drowsy Driving Prevention Using Image Recognition and IoT

**Seok-Woo Jang** [1] **and Byeongtae Ahn** [2,*] 

[1]    Department of Software, Anyang University, Anyang 14028, Korea; swjang@anyang.ac.kr
[2]    Liberal and Arts College, Anyang University, Anyang 14028, Korea
*    Correspondence: ahnbt@anyang.ac.kr

**Abstract:** In recent years, the casualties of traffic accidents caused by driving cars have been gradually increasing. In particular, there are more serious injuries and deaths than minor injuries, and the damage due to major accidents is increasing. In particular, heavy cargo trucks and high-speed bus accidents that occur during driving in the middle of the night have emerged as serious social problems. Therefore, in this study, a drowsiness prevention system was developed to prevent large-scale disasters caused by traffic accidents. In this study, machine learning was applied to predict drowsiness and improve drowsiness prediction using facial recognition technology and eye-blink recognition technology. Additionally, a $CO_2$ sensor chip was used to detect additional drowsiness. Speech recognition technology can also be used to apply Speech to Text (STT), allowing a driver to request their desired music or make a call to avoid drowsiness while driving.

**Keywords:** drowsy; driving; prevention; detection; real-time flicker recognition method

## 1. Introduction

According to the statistics of Korea on traffic accidents in the last 5 years, driving while drowsy is one of the most important factors in traffic accidents, and its related mortality rate is more than 2 times higher than other causes of traffic accidents [1]. As a solution to resolve these problems, it is possible to reduce the mortality rate of such traffic accidents by detecting and preventing drivers from driving while drowsy. Therefore, studies aimed at detecting and preventing this kind of driving have been actively researched in the academic field [2,3]. In this study, a device for preventing drowsy driving was selected, and interviews and surveys were conducted with operators who do a large amount of driving. The survey consisted of a total of 61 questions related to car driving and lifestyle, driving habits related to drowsy driving, the use of peripheral devices, the vehicle environment, accidents, and drowsy driving in order to gain insights to refine our ideas. The survey was conducted with about 200 people. The survey was conducted online.

Figure 1 shows the classification of brainstorming solutions through the survey.

Three techniques are used to detect the drowsiness of commercial drivers: recognizing the driver's eyes through cameras and using biosignals such as breathing, temperature, and heart rate to analyze operation patterns, such as the abnormal use of pedals and steering wheels.

Among the existing third-party products, the drowsiness detection system through the use of a camera has a low accuracy and a high error rate. In addition, various methods, such as detecting breathing irregularity, temperature rise, heart rate irregularity, etc., also has poor accuracy and a high error rate. In order to increase the detection of drowsiness, the proposed method in this paper improves accuracy through face detection as well as eye-blink detection and further improves drowsiness prevention by using a carbon dioxide sensor chip. Therefore, the system proposed in this paper improves the performance of the drowsiness prevention system by increasing the accuracy

and reducing the error rate of drowsiness detection by combining several methods of the existing products used to detect and prevent drowsy driving. Additionally, to prevent drowsiness, music and broadcasting are automatically turned on or the driver can select music and broadcasting by using their voice. The details will be discussed in detail below. This study reviews current drowsiness detection technology through domestic and foreign cases in Section 2 and proposes a design to detect drowsy driving by introducing a $CO_2$ sensor chip and checking the blink rate of the eyelids in Section 3. In Section 4, a system built based on the design of Section 3 is presented. In Section 5, a conclusion and future research are proposed.

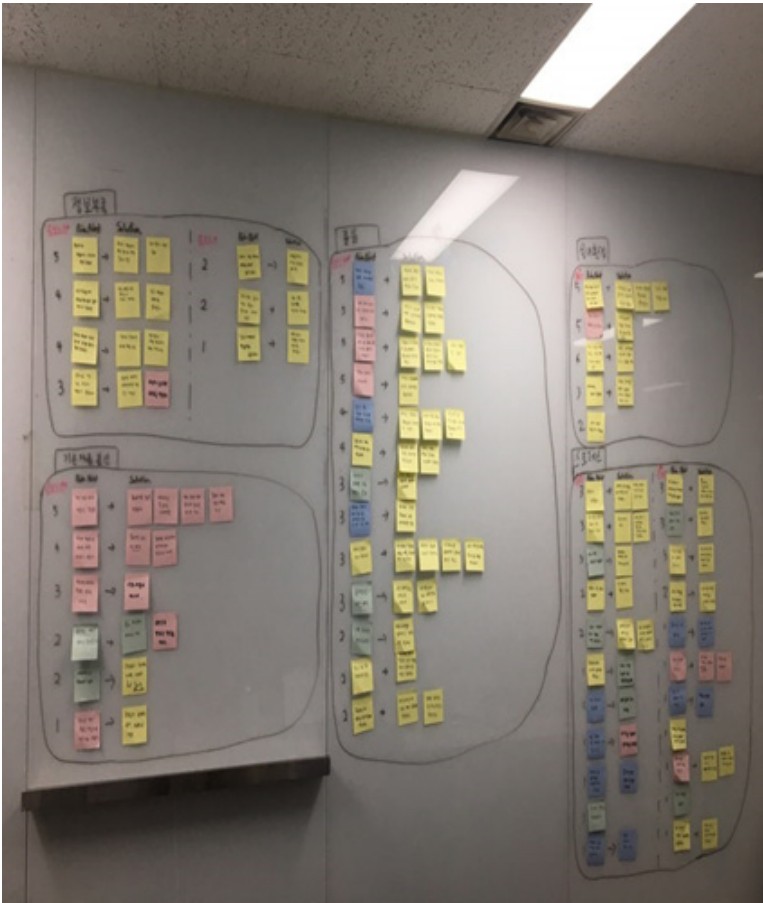

**Figure 1.** Solution of brainstorming on the survey.

## 2. Cases

To detect and prevent drowsy driving, we proposed overseas and domestic cases. The cases investigated were systems from Lexus, Ford, and Hyundai Mobis, respectively.

### 2.1. Domestic Cases

The Hyundai Mobis system has been the most actively studied as a system for detecting and preventing drowsiness while driving domestic cars. This system prevents drowsy driving by recognizing the driver's face and detecting breathing conditions [4]. As a function of the system, the beep sounds when the driver's eyes do not stare to the front. It is designed to prevent driving during drowsiness, and an alarm is activated by checking the blink time and speed (cycle) of the eyelid [5,6]. However, it is still in the design phase and has not been developed.

Figure 2 is a sleepiness prevention system under development for use by Hyundai Mobis. Surveillance cameras detect the driver's eyes, nose, and mouth and continuously provide statistical data [7].

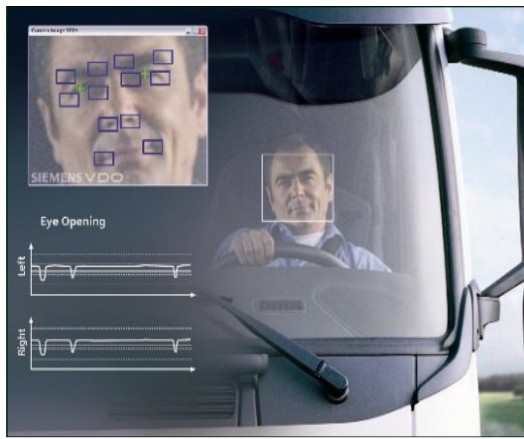

**Figure 2.** Mobis by Hyundai.

Other domestic car manufacturers have developed sensors to detect gaps between cars. When the distance between the sensor and the vehicle in front is narrow, an alarm is provided to the driver and the brake is forced to operate [8]. However, there is no drowsiness detection and prevention system in Korea that supports all of these factors.

In Figure 3, the front sensor detects the lane of the vehicle and sounds an alarm to warn the driver when the vehicle leaves the lane [9,10]. This capability is currently being added to all vehicles.

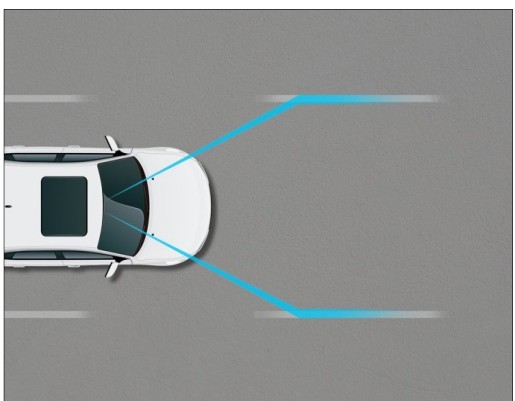

**Figure 3.** Camera sensors on the car.

Figure 4 measures the tilt of the driver's face with a tilt and vibration sensor on the ear flap and generates a vibration when a certain tilt value is detected [11]. Currently, no product has applied this function. However, it could be applied to prevent drowsy driving in the future.

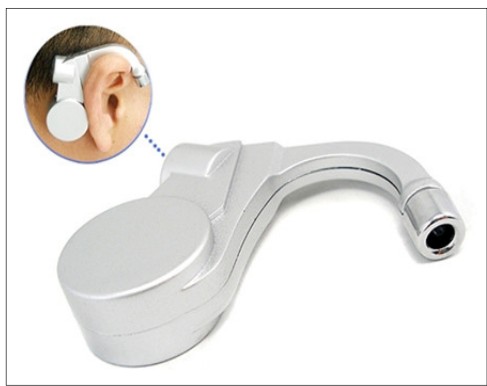

**Figure 4.** Inclination detection vibration sensor.

## 2.2. Overseas Cases

Toyota's Lexus vehicles have applied various methods to prevent drowsy driving. The Lexus model, launched in 2008, is equipped with Toyota's drowsy driving protection. Figure 5 shows an infrared camera attached behind the steering wheel to recognize a driver's drowsiness and sound a warning [12].

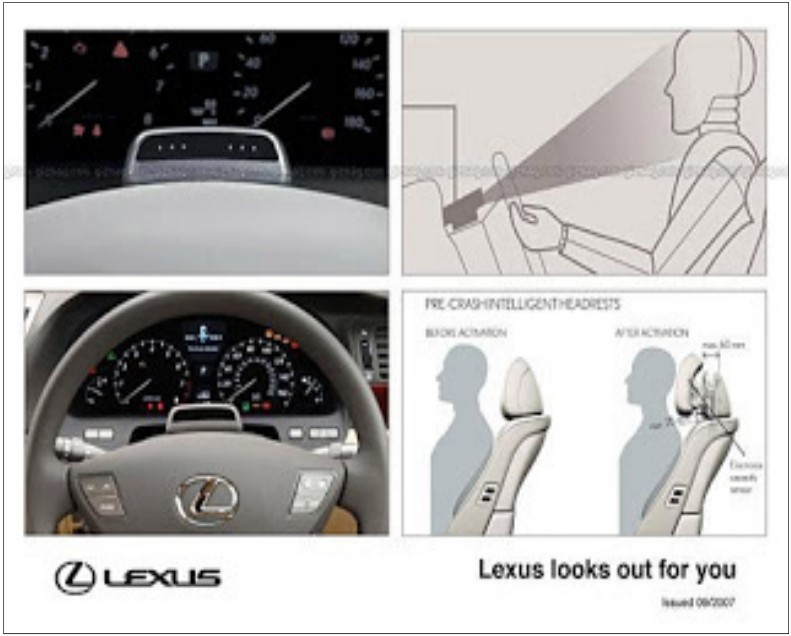

**Figure 5.** Drowsy driving prevention by Lexus.

Toyota's Lexus vehicles employ a variety of methods and techniques to prevent drowsy driving. However, they are still in the design phase and development has not been completed. However, a commercially available function is the camera recognition function.

(1) Image processing technology that recognizes the driver with a camera is used, modeled in 3D with 238 points and 913 meshes.

(2) Information about the driver's emotions is extracted using mesh points.

(3) The analysis system recognizes the driver's behavior, such as looking at another person or looking at a smartphone.

(4) Hundreds of factors are used to increase accuracy for a variety of analytical awareness skills, such as age, gender, and race.

(5) The driver's eyes are calculated by analyzing the distance between the eyelids and the eyelids. A total of four alarm systems are used to identify the risk of an accident and provide different situational information for each alarm step.

Figure 6 shows the technology developed by Ford to detect and prevent drowsy driving. Ford has applied the following techniques to prevent drowsy driving [13]. Ford's system uses an infrared sensor on the handlebar, a heart-rate sensor, and a breathing sensor on the seat belt to collect biometric information. Then, after analyzing the steering wheel and pedals, the driver's fatigue is calculated to support an alarm sound. Ford's vehicles have a temperature sensor and heart rate detection. However, the accuracy of drowsy driving detection is not high with these features alone.

- Group analysis is performed using the vehicle information (acceleration, length, slope).
- The driver's driving activity (acceleration pedal and brake pedal tilt), surroundings (road and traffic conditions), and the driver's biometric information (temperature, body temperature, breathing rate, and heart rate) are determined to check their condition.

•    Heart rate is monitored using six electronic sensors in the driver's seat.

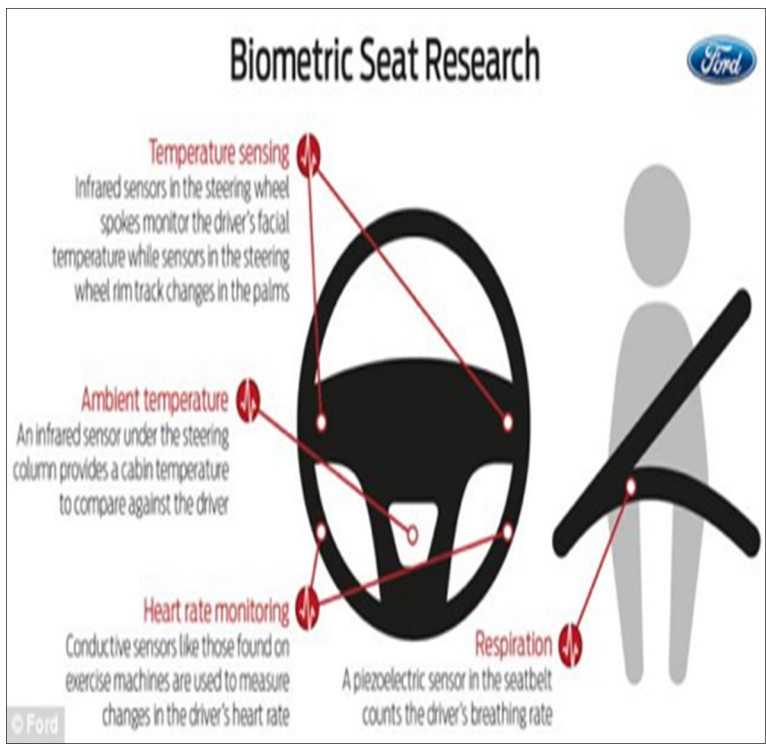

**Figure 6.** Drowsy driving prevention by Ford.

## 3. Design of Drowsiness Detection

### 3.1. Eye-Blink Detection

Face recognition should be first performed in order to detect eye blinking. Therefore, the system recognizes the pupils of the driver's eyes after recognizing the face and examines the blink speed of their eyelids to detect drowsiness.

In Figure 7, the Haar Cascade technique, which uses patterns of light and shade in OpenCV, is applied to recognize the faces of human beings. On drivers' faces, the eyes are dark and the nose is bright. Therefore, the technique extracts face information by analyzing the pattern in the black and white image. Further, extracted face information is recognized by using the Haar Cascade in OpenCV [14]. The study actually applied this technique to recognize the eyes of a human face. However, the Raspberry Pi environment used in this study was poor in terms of its performance. Thus, this study referred to Korean standard face data to determine the eye position of drivers.

Figure 8 shows that the eyes of drivers can be continuously tracked by applying the mean-shift method to continue to track them even when they move. The mean-shift is a method used to find the peak or center of gravity of data distribution, which indicates the algorithm is moving to a data-dense area and the center of the distribution. When the data are distributed on a 2D plane, the process of finding the densest peak point of the data is constructed by the following methods [15]:

(1)　Obtain data originating from the radius, r, from the current position.
(2)　Move the current position to the coordinates of the center of gravity.
(3)　Repeat step 1 and 2 until the position converges.

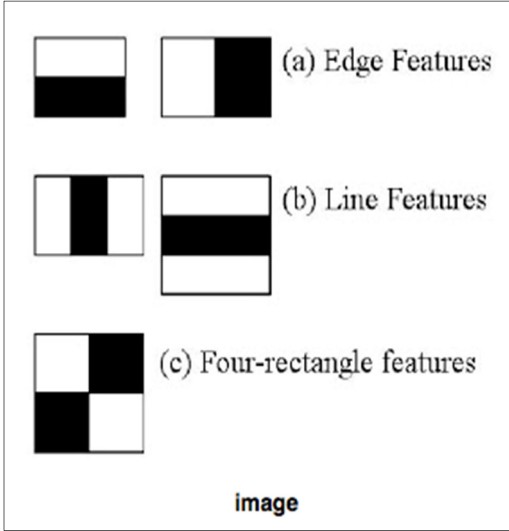

**Figure 7.** Haar Cascade method.

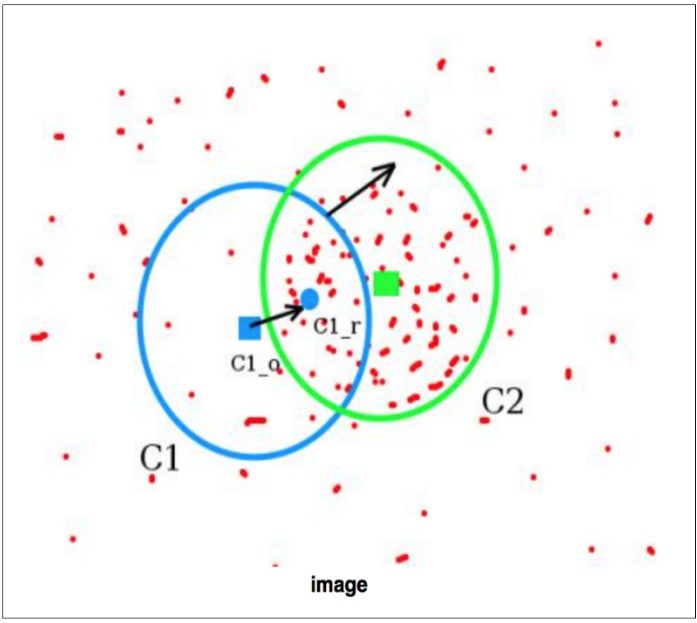

**Figure 8.** Mean-shift method.

For pupil detection, OpenCV's Hough Circle Transform was applied. This method detects pupils in the eye region. The detected pupils are binarized and filtered; all pixels brighter than the threshold are designated as white, and the other areas are designated as black. The following formula was applied to obtain a binary image:

$$\mathrm{T}(x, y) = \frac{1}{n^2} \sum_{x_i} \sum_{y_i} I\left(x + x_i, y + y_j\right) - C \tag{1}$$

Then, using the Canny edge detection algorithm, an object boundary was found in the image. This method is efficient for contour extraction and can remove all contours related to gray matter in the original image. This technology improves upon the existing face-recognition and blink-detection technology, and machine learning was also applied to improve detection performance. The existing pure image processing could not achieve the desired performance within a short period of time. Therefore, it was decided to create a program to learn what the state of the new frame would be after

determining the position of the eyes and the open eyes. By creating a module that stored the image of the user's eye area in the category.number.jpg format, a vector with 1024 features was created. This value was determined through a hyperparameter tuning experiment. Softmax was used as the activation function. Stockastic gradient descent was used as the learning model.

### 3.2. Carbon Dioxide Detection

As a result of a questionnaire survey, it was found that the occurrence of many drowsy driving operations depend on the air quality in vehicles. Therefore, this study tried to prevent drowsy driving by detecting the concentration of carbon dioxide in vehicles.

Figure 9 represents a sensor for measuring the concentration of carbon dioxide of the NDIR(Non-Dispersive Infrared) system. If the concentration of carbon dioxide was over 1500 ppm, it was expected that drowsiness would appear. Further, when the concentration of carbon dioxide was high, this not only caused drowsiness and stiffness but also caused dizziness, headache and health problems. This sensor measures the concentration of carbon dioxide to the extent that refraction is caused by gas concentration using a non-distributed infrared emitting unit. The sensor has high durability and high accuracy, thereby detecting drowsy driving quickly. From the possible semiconductor resistance, electrochemical, and NDIR sensors, this study used the NDIR method to measure driving conditions, due to the considerations of cost-effectiveness and efficiency.

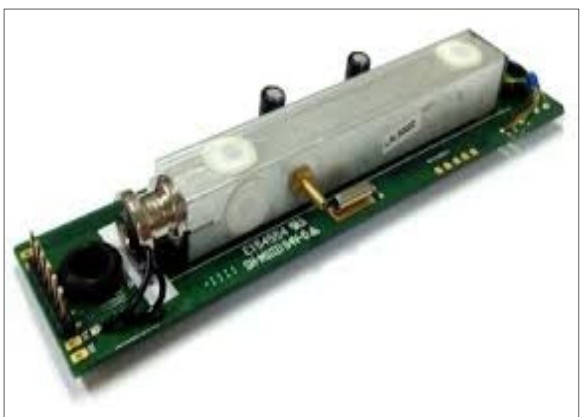

**Figure 9.** Carbon dioxide concentration sensor.

### 3.3. System Configuration Diagram

The configuration diagram of the drowsy driving detection and prevention system in this study mainly consists of three stages. The first step is an input part and consists of sensors, microphones, and cameras. The second step is an internal module part and consists of Speech to Text (STT), drowsiness prevention, internal environments, and entertainment [16].

The final step is part of this project, and its system configuration diagram represents the whole configuration diagram of this project.

Figure 10 represents the system configuration diagram. The following items summarize the system configuration diagram: The project name of this system is JARVIS. Carbon dioxide sensor chips and camera images are used to detect drowsiness in the vehicle. The camera detects face recognition and blinking and automatically turns on music and broadcasting when drowsiness is detected. The driver can request a broadcast or desired music through the microphone. Because cameras and sensors are built into the diffuser, they are not spatially and visually constrained. To take a break, the driver can use a mobile to guide them to nearby rest areas, and this information is provided in the database.

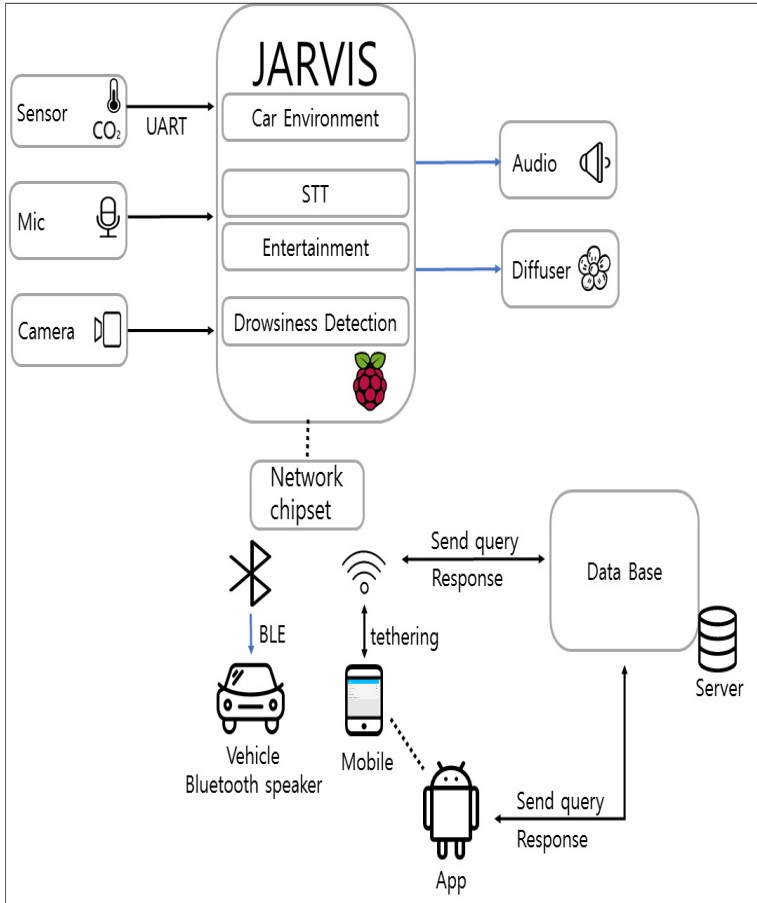

**Figure 10.** Structure of the system.

i.   Input

- Sensor: $CO_2$ temperature sensor, which can measure the concentration of carbon dioxide up to 3000 ppm.
- Microphone: microphone that receives the user's voice commands.
- Camera: infrared camera that can operate in the night.
- Internal Module.
- STT: infrared camera that can operate in the night.
- Drowsiness detection: drowsiness detection module.
- Internal Environment: temperature and carbon dioxide sensor, module, and weather and environment module.
- Entertainment: broadcasting, music, and news module.

ii.  Output

- BLE: output by a built-in car speaker with a Bluetooth connection.
- Audio: motion output of automobile secretary (audio guidance, conversation, broadcasting, and music).
- Air freshener: operation for indoor ventilation and drowsiness prevention.

## 4. Implementation of Prevention System

Figure 11 indicates some of the actual developed sources. There were many cases of incorrect detection, such as eyebrows, hair, etc. Therefore, this study implemented this source for reducing

incorrect detection by exactly recognizing the left eye and right eye when both eyes are located in each facial position according to the center of the face.

```python
left_eye = right_eye = None

if len(eyes) > 0:
    for i, v in enumerate(eyes):
        ex = v[0]
        ey = v[1]
        ew = v[2]
        eh = v[3]
        if ex + (ew/2) <= w / 2:
            left_eye = eyes[i]
        elif ex + (ew/2) > w / 2:
            right_eye = eyes[i]

    if left_eye is not None:
        cv2.rectangle(img, (x + left_eye[0], y + left_eye[1]),
                          (x + left_eye[0] + left_eye[2], y + left_eye[1] + left_eye[3]), (0, 255, 0), 2)

    if right_eye is not None:
        cv2.rectangle(img, (x + right_eye[0], y + right_eye[1]),
                          (x + right_eye[0] + right_eye[2], y + right_eye[1] + right_eye[3]), (255, 255, 0), 2)
```

**Figure 11.** Example of the source.

Figure 12 shows the product modeling implemented in this paper. The hard appearance of electronic products was avoided, and Kakao Friends were used due to the value of their friendly and functional design, making them suitable for female users and as gifts.

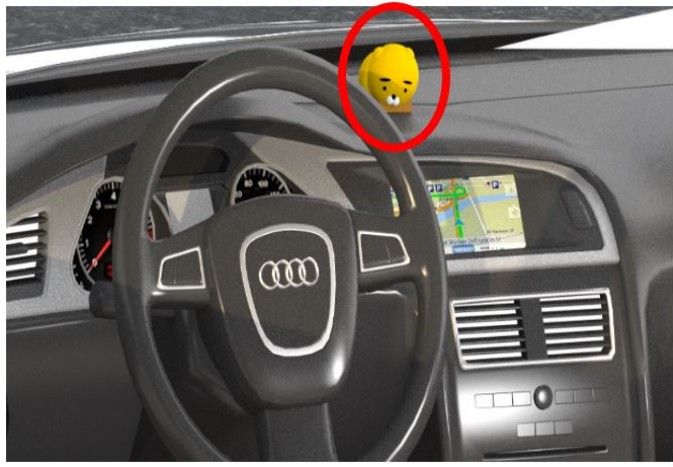

**Figure 12.** Image of the completed product.

Figure 13 shows a scene that detects blinking. The results from the Raspberry Pi for carbon dioxide detection alone made it difficult to detect drowsy driving. Therefore, it was possible to improve the prevention of drowsy driving by improving the algorithm for image processing as much as possible and efficiently detecting eye blinking using deep learning [17].

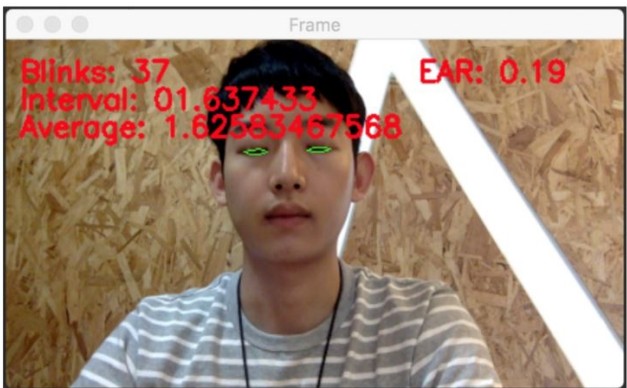

**Figure 13.** Shot of eye-blink detection.

A voice recognition API (Application Program Interface) is required to support voice services. Speech recognition support products include CMU Sphinx, IBM, Google Cloud Speech, Bing Speech, wit.ai, Google Speech, and Houndify.

Figure 14 shows the product developed in this paper. The camera detects blinking eyes, and the carbon dioxide sensor inside the box detects carbon dioxide in the car and automatically ventilates it. In this system, this was developed as a fragrance, and music was automatically played when drowsiness was detected.

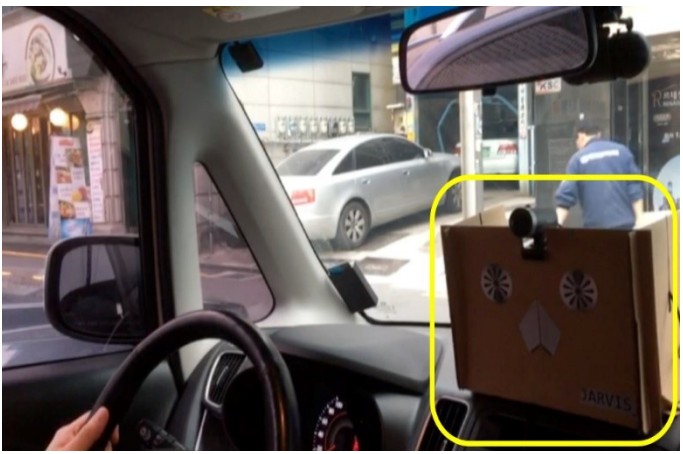

**Figure 14.** Image of the developed product.

Figure 15 shows the various functions of the drowsy driving prevention system. In the default settings, you can configure settings for the account, drowsy driving detection, convenience functions, and Bluetooth. The drowsy driving detection includes built-in driving-time alerts, drowsiness sensitivity figures, air fresheners, stretching recommendations, and nighttime teaming. The driving-time reminder can check fatigue levels by displaying the driving time every hour. The drowsiness sensitivity figures automatically play music when the number exceeds 60 and guide you to the nearest rest area. Nevertheless, if drowsiness is detected, stretching is recommended in the announcement. In the convenience function, various functions, such as music, current news, and humor, are provided according to the drowsy driving detection value, and the temperature and humidity in the vehicle are checked and provided periodically.

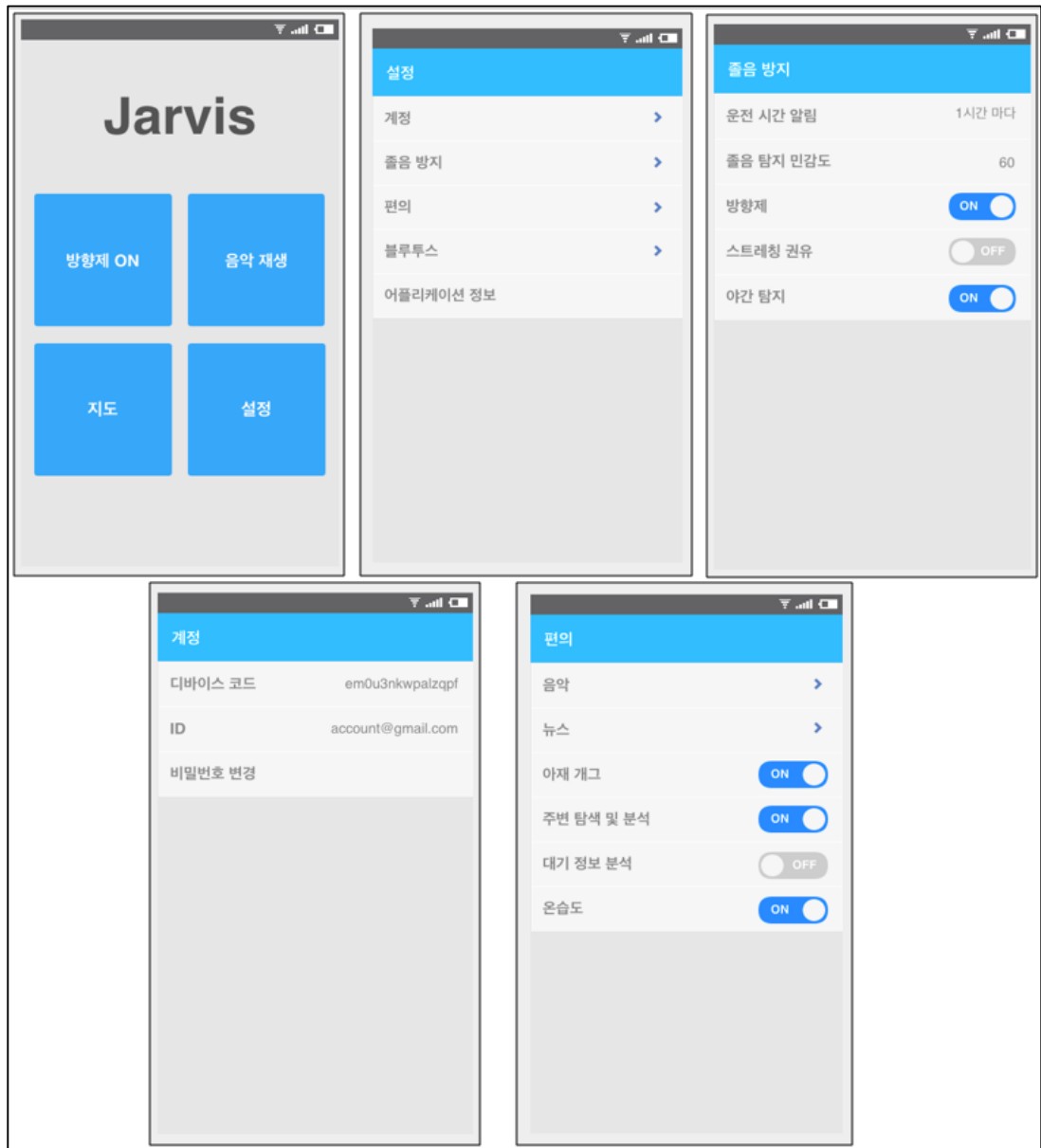

**Figure 15.** Function of the drowsy driving prevention system.

## 5. Conclusions and Future Works

This study designed a system to detect and prevent drivers from driving while drowsy. In addition, it developed and tested actual cases based on the design. It used Python and C languages for its developmental environment, and it used a Raspberry Pi 3, infrared camera, speaker, microphone, carbon dioxide sensor, Galaxy S4, and the automobile model Sonata. In the future, we plan to build a system that is linked with an application on a smartphone and check the real-time reaction rate through actual testing.

Table 1 compares and analyzes the functions of this system with domestic and foreign products that are currently commercialized.

Finally, we will develop an application not only on Android but also for an iOS environment for further development and inter-working.

**Table 1.** Comparison analysis of the product.

|  | Hyundai | Toyota | Ford | Our System |
|---|---|---|---|---|
| **Face detection** | O | O | O | O |
| **Eye-blink detection** | X | O | O | O |
| **IoT sensor chip ($CO_2$)** | X | X | O | O |
| **Auto (TV, radio) play** | X | X | X | O |
| **Rest-area map position information** | X | X | X | O |
| **Auto air fresh** | X | X | X | O |
| **Heart rate check** | X | X | O | X |

**Author Contributions:** The first author contributed to conceptualization and image recognition. Corresponding author contributed to system design, implementation, review and editing. All authors have read and agreed to the published version of the manuscript.

**Funding:** This research received no external funding.

**Conflicts of Interest:** The authors declare no conflict of interest.

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
