# Peer review of "Implementation of Detection System for Drowsy Driving Prevention Using Image Recognition and IoT"

_sustainability, doi:10.3390/su12073037_

Round 1

Reviewer 1 Report

Review of the manuscript # sustainability-743545-peer-review-v1
Title: Implementation of Detection System for Drowsy 2 Driving Prevention using Image Recognition and IoT

Comments / Feedback

The manuscript deals with an exciting and real world problem of drowsy driving. The authors have done a good job on elaborating different parameters to address the above said issue. Overall, the paper is average and hardly demonstrates any merit and novelity. Nonetheless, following suggestion shall be taken into account before publication in the journal.

  1. Abstract, Introduction and literature review
      a. Need to add methodology in a more elabortive way and also discuss implications in the abstract.
      b. There is a need to expend the introduction and highlight the research gaps and how this paper contributed in overcoming those gaps.
      c. In introduction section, I would suggest adding a clear research question or hypothesis based on the discussed literature and research gap.
      d. In literature review there is an overwhelming amount of figures, which makes it look more like a report not a scientific study, I would suggest to delete pictures, figure 1, 3, 6, 8, 11, does not add much value to the manuscript, consider deleting them. Where as remaining pictures must be atleast 600 dpi so that they are clear and readable.
      e. It would be nice to include some scenarios or examples to make reader understand the real world challenges.
  2. It might be a good idea to look at the following papers for theory building:
      a. Azmat, M.; Kummer, S.; Moura, L.T.; Gennaro, F.D.; Moser, R. Future Outlook of Highway Operations with Implementation of Innovative Technologies Like AV, CV, IoT and Big Data. Logistics 2019, 3, 15.
      b. Wintersberger, S.; Azmat, M.; Kummer, S. Are We Ready to Ride Autonomous Vehicles? A Pilot Study on Austrian Consumers’ Perspective. Logistics 2019, 3, 20.
      c. Azmat, M., Kummer, S. Potential applications of unmanned ground and aerial vehicles to mitigate challenges of transport and logistics-related critical success factors in the humanitarian supply chain. AJSSR 5, 3 (2020). https://doi.org/10.1186/s41180-020-0033-7
      d. Michail, M.; Konstantinos, M.; Biagio, C.; María, A.R.; Christian, T. Assessing the Impact of Connected and Automated Vehicles. A Freeway Scenario. In Advanced Microsystems For Automotive Applications 2017; Springer: Cham, Switzerland, 2018
  3. The paper’s methodology seems to be sound. However, it hasn’t been elaborated enough and no proper citations and referencing is made to understand why such an approach has been used.
      a. No information has been provided on how many vehicles were used in the application testing and how was it tested. It is important mention sample size and how and why certain techniques were used and how the test group was selected.
  4. It is highly recommended to add research limitations and implications of the findings. I would suggest these to be added in the conclusion and also briefly in the abstract. It would be also nice to see some potential applications scenario or examples for the real world use.
  5. In general, the use of the English language is ok. However, proofreading is required; especially, the sentence structure could have been better on several occasions.
    I wish the authors good luck with the revision and resubmission.

Reviewer 2 Report

The paper adds a new perspective on drowsiness prevention systems and is highly practical. Content has merit, but accuracy, clarity, completeness and writing should be improved in time.

Some of the many inconsistencies noticed are:

27-30: the three techniques mentioned seem to be only 2; "Figure 1 presents that the drowsiness prevention system" ???; 50: "The image processing technology, recognizes drivers with a camera, is used"; 87-88 and 89-90 // 91-92, 93-94 are duplicated!; 150 ?"As a result of a questionnaire survey" - no metnion about the source of the survey results" etc.

Also, please explain the 76% similarity rate identified by Turnitin plagiarism prevention solution.

Reviewer 3 Report

In this paper, the authors claimed that they developed a drowsy driving detection and prevention system. Overall, the reviewer felt the paper is too simplistic to be suitable for publication on an academic journal paper. It’s read more like a story in a newspaper or magazine. It is hard to see any academic merit or contribution from this current presentation of this paper. The reviewer is highly encouraging authors to improve the manuscript especially in terms of readability: Clearer explanations, more careful writing and a more schematic approach are required in the opinion of the reviewer to consider this manuscript for publication. Here are some specific concerns of the reviewer:

  • For all figures in this paper, can the authors specify which ones are created by themselves and which ones are cited from other resources? This is important due to a plagiarism concern.
  • The abstract is not clearly and too simple. Need to rewrite and highlight the contributions of this study. For example, the authors mentioned that “more than 70% of major accidents occur in drowsy driving”. Any data to support this statement? The authors also claimed their system has improved the efficiency of existing techniques. But how?
  • Line 31, “low accuracy and low error rate…” you mean “high error rate”??
  • Chapter 2 reads more like a literature review part. Please enhance this part and make it like a sound and complete literature review section.
  • Lines 82-98: some words are duplicated.
  • For some figures (e.g., Figure 11), please translate Korean texts into English, considering the reader is from all of the world.
  • For Figure 13, I don’t understand this structure. Please elaborate on it.
  • Is this system a real one or still under development? Since in the conclusion part, the authors mentioned the future is to build this kind of system for a real-time application, which makes the reviewer feel like this system is still under development.

Round 2

Reviewer 1 Report

Dear Authors,

The revision looks much better and will add to the research in the smart ways of overcoming drowsiness. However, i will suggest the following before publication of this article.

please add the high-resolution pictures, several pictures are not clear and some are oversized. Reduce the number of images in the manuscript you do not need to insert a picture with each example.

all the best.

Author Response

Thank you very much.

Reviewer 2 Report

I noticed important improvements compared to the first version of the paper. The presentation is significantly clearer. Still, the plagiarism problem persists - a 49% in Turnitin. Please solve this and, for future manuscripts, please provide them in an honorable English version.

Author Response

Thank you very much.

Reviewer 3 Report

Thanks for the efforts of the authors and the reviewer can see significant changes throughout the revised manuscript. However, there are still lots of gaps to be filled in. Please see the comments below:

  • In the abstract, there are still some sentences read awkwardly. For example:

  • “and serious accidents related to serious injuries and deaths are increasing more than for the general public”

  • “and prediction was improved by applying machine learning to improve 16 drowsiness prediction.”

I also find this kind of sentences throughout the paper. These are not acceptable and please revise them.

  • Figure1: Better to make a schematic drawing rather than this kind of picture. Can you also elaborate on the results from the survey and how you use the results in your study (especially the design process of your system)?
  • In Section 2, Figure 10: What does C1 or C2 stand for? What does C1_o or C1_r stand for? What does the red dot stand for? Any pseudo-codes for the mean-shift method? You’d better number any equation in your text (for example, Page 9, Line 147).
  • Page 17, Line 167, what is “NDIR”?
  • Page 11-12, Lines 194-209: You’d better use different bullets for different list levels
  • In Section 4: Can you provide any results or data from your implementation? Like, how many times for your system to detect the driver’s drowsy behavior? What is the efficiency of your system?

Overall, the quality of this manuscript is still below the reviewer’s expectation and the reviewer is still not favorable for the publication of this manuscript.  

Author Response

Thank you very much.

Round 3

Reviewer 3 Report

Thanks for the efforts of the authors and the reviewer can see significant changes throughout the revised manuscript. However, there are still lots of gaps to be filled in. Please see the comments below:

In the abstract, there are still some sentences read awkwardly. For example:

  • “and serious accidents related to serious injuries and deaths are increasing more than for the general public”
  • “and prediction was improved by applying machine learning to improve 16 drowsiness prediction.”

I also find this kind of sentences throughout the paper. These are not acceptable and please revise them.

  • Figure1: Better to make a schematic drawing rather than this kind of picture. Can you also elaborate on the results from the survey and how you use the results in your study (especially the design process of your system)?
  • In Section 2, Figure 10: What does C1 or C2 stand for? What does C1_o or C1_r stand for? What does the red dot stand for? Any pseudo-code for the mean-shift method? You’d better number any equation in your text (for example, Page 9, Line 147).
  • Page 17, Line 167, what is “NDIR”?
  • Page 11-12, Lines 194-209: You’d better use different bullets for different list levels
  • In Section 4: Can you provide any results or data from your implementation? Like, how many times for your system to detect the driver’s drowsy behavior? What is the efficiency of your system?

Overall, the quality of this manuscript is still below the reviewer’s expectation and the reviewer is still not favorable for the publication of this manuscript.